**Data Availability Statement:** This study reports on qualitative data which by nature is difficult to fully anonymize. Participants were not asked to agree to

# Filling the GAP: Integrating a gambling addiction program into a shelter setting for people experiencing poverty and homelessness

**Flora I. Matheson**[1,2]*, **Sarah Hamilton-Wright**[1], **Tara Hahmann**[1], **Arthur McLuhan**[1], **Guido Tacchini**[1], **Aklilu Wendaferew**[3], **Parisa Dastoori**[1,2]

**1** MAP Centre for Urban Health Solutions, St. Michael's Hospital, Toronto, Ontario, Canada, **2** Dalla Lana School of Public Health and Centre for Criminology & Sociolegal Studies, University of Toronto, Toronto, Ontario, Canada, **3** Good Shepherd Ministries, Toronto, Ontario, Canada

* Flora.Matheson@unityhealth.to

## Abstract

The burden of harm from problem gambling weighs heavily on those experiencing poverty and homelessness, yet most problem gambling prevention and treatment services are not designed to address the complex needs and challenges of this population. To redress this service gap, a multi-service agency within a shelter setting in a large urban centre developed and implemented a population-tailored, person-centred, evidence-informed gambling addiction program for its clients. The purpose of this article is to report on qualitative findings from an early evaluation of the program, the first designed to address problem gambling for people experiencing poverty and/or homelessness and delivered within a shelter service agency. Three themes emerged which were related to three program outcome categories. These included increasing awareness of gambling harms and reducing gambling behaviour; reorienting relationships with money; and, seeking, securing, and stabilizing shelter. The data suggest that problem gambling treatment within the context of poverty and homelessness benefits from an approach and setting that meets the unique needs of this community. The introduction of gambling treatment into this multi-service delivery model addressed the complex needs of the service users through integrated and person-centered approaches to care that responded to client needs, fostered therapeutic relationships, reduced experiences of discrimination and stigma, and enhanced recovery. In developing the Gambling Addiction Program, the agency drew on evidence-based approaches to problem gambling treatment and extensive experience working with the target population. Within a short timeframe, the program supported participants in the process of recovery, enhancing their understanding and control of their gambling selves, behaviours, and harms. This project demonstrates that gambling within the context of poverty requires a unique treatment space and approach.

their data being available for other studies, as they experience significant social and health burdens that are highly stigmatized, including problem gambling, homelessness, poverty, and a range of complex social and health challenges. Requesting that they agree to share their data more publicly would have deterred some from taking part in the study and would have the potential to significantly alter the data obtained. Ethical approval for this research study was granted by St. Michael's Hospital Research Ethics Board [REB 17-058] on the basis that participants' data was only accessible by the research team. The authors believe in the principle of making data freely available but this was not anticipated or specifically approved by the St. Michael's Hospital Research Ethics Board at the time the study was initiated. However, specific requests for data sharing for purposes such as data verification and meta-analysis will be considered by the St. Michael's Hospital Research Ethics Board on an individual basis under defined and mutually agreed-upon conditions. Requests for access to the data may be directed to the Research Ethics Board, researchethics@smh.ca or Sharon. Freitag@unityhealth.to.

**Funding:** The project was funded by the Ontario Government and the Ontario Trillium Foundation [LP95159] (AW, FIM). The funders had no role in study design, data collection and analysis, decision to publish, or preparation of the manuscript. https://otf.ca/; https://www.ontario.ca/.

**Competing interests:** The authors have declared that no competing interests exist.

## Introduction

Problem gambling (PG) is associated with an array of complex social problems and public health issues [1,2]. For example, PG is associated with multiple measures of poverty, including low income [3–5], precarious employment [6,7], and unemployment [4,8,9]. PG is also associated with housing instability and homelessness [10,11]. The prevalence of PG among people experiencing homelessness is striking, with rates as high as 58% [4].

The relationship between PG, poverty, and housing instability is a significant and growing public health concern [1,12,13]. Research has documented multiple comorbidities and challenges among people experiencing PG, homelessness, and poverty [3,4], including mental illness, substance use, criminalization, poor coping skills, relationship loss, employment instability, financial insecurity, experiences of discrimination and stigma [2], and histories of trauma [14]. Effective responses to complex co-occurring health and social issues among this population require awareness of the relationship between PG and poverty/homelessness, widespread screening for PG among service agency clients, as well as population-targeted and agency-coordinated PG prevention and recovery services [1,2,11,15].

However, most PG prevention and recovery services are not tailored to the particular needs and challenges of some of the communities with the greatest prevalence of PG, and no studies to date have evaluated a PG intervention for people experiencing homelessness and poverty [16]. For example, a recent study of PG among clients of a multi-service agency within a shelter setting located in downtown Toronto, revealed a significant gap in PG services for this population [17]. Although some psychological treatments show promise among people experiencing PG more generally [18]—such as cognitive behavioural therapy programs on their own [19,20] or combined with other treatments [21–23]—the complex needs of people experiencing PG, homelessness, and poverty may reduce the effectiveness or accessibility of standard treatment options [1,2].

Shelter service agencies, a first point of contact for those contending with poverty and homelessness, are well-positioned to implement PG services that address the needs of their clients. Good Shepherd Ministries (GSM), a service agency that provides shelter-based coordinated care to adults with unmet health and social needs, developed and implemented the Gambling Addiction Program (GAP) in April 2017. The purpose of this article is to report on the results of an early evaluation of the GAP. To our knowledge, this is the first treatment program designed to address PG within the context of poverty and homelessness and offered within a shelter service agency. First, we overview the data and methodological approach of the study, including a description of the setting and PG intervention. Second, we present GAP client and case worker perspectives on program delivery and program impact. We conclude by discussing the implications and limitations of the study as well as future directions for research.

## Methods

This article is part of a larger mixed methods evaluation [24] of a gambling treatment program within a shelter service agency. The larger project employed a community-based participatory approach [25,26], which pursues "systematic inquiry, with the collaboration of those affected by the issue being studied, for the purposes of education and taking action or effecting social change" [25, p. 1927]. We have worked collaboratively with our partner, GSM of Toronto, throughout the study, from design to dissemination. Consistent with participatory action research methods [27], the research team conducted the analysis independently from the partner agency staff, but we worked together to develop the recommendations from the findings.

## Ethical considerations

The study followed the principles of the Tri-Council Policy Statement: Ethical Conduct for Research Involving Humans [28]. Participants provided informed verbal consent prior to the interviews. Confidentiality was ensured by assigning each participant a study identification number on all study materials. We obtained verbal, rather than written consent, to offer an added layer of protection for participants who experience stigmatized identities. The project, including the verbal consent procedure, was approved by the Research Ethics Board [REB # 17–058] of St. Michael's Hospital, Toronto, Ontario, Canada. Verbal consent was recorded in the consent form and on the recruitment database.

## Good Shepherd Ministries (GSM)

GSM is a service agency situated in the inner city of Toronto, Ontario, Canada. The agency provides compassionate care to those in need, connecting clients with housing, medical care, and counselling. GSM offers a suite of services including drop-in and meal programs; clothing services; short stay addictions pre- and post-treatment programs; shelter services; medical, psychiatric and addictions clinics; supportive housing programs for the elderly and those living with HIV/ AIDS; pastoral care, social enterprise/training, employment support; and housing resettlement case management. The clientele is approximately 90% male.

## Gambling Addiction Program (GAP)

GSM designed the GAP as a program to reduce gambling harms among clients experiencing poverty, homelessness and multimorbidity. Two Gambling Addiction Case Workers (GACWs) provided individual case management and facilitated group sessions in Cognitive Behavioural Therapy (CBT) and Life Skills (LS). Another agency provided trusteeship services to help clients manage their money. GSM also partnered with Gamblers Anonymous (GA) to offer weekly onsite meetings as an additional resource available to GAP clients, the broader GSM community, and clients of other service agencies. GA operated independently from the GAP, and therefore fell outside the formal scope of this evaluation.

GSM nested the GAP within its broader suite of services available to clients. Clients could access the program through either internal referrals or external agency referrals. Registration for overnight shelter beds provided an opportunity for internal referrals. Staff asked clients if they would voluntarily answer the NODS CLiP, a 3-question screener to identify people at risk of PG [29], and referred those who answered at least one question affirmatively to a GACW. GACWs performed outreach at other community agencies serving the target population that provided the basis for external referrals to the program. Internal and external referrals connected individuals with a GACW for the initial case management intake, where the GACWs administered the Canadian Problem Gambling Severity Index (PGSI) [30,31]. The enrolment of clients into the program was based on scoring 3 or more on the PGSI. A score of 3 to 7 indicates moderate level of problems leading to some negative consequences and a score of 8 or more indicates problem gambling with negative consequences and a possible loss of control.

## Recruitment

GACWs introduced the study to each client during GAP intake. Clients were eligible to enroll in the research study if they agreed to participate in the GAP. If a client agreed to participate, a GSM staff member, trained by the research team and outside the circle of care, explained the study in further detail and conducted the informed consent process. Participants provided informed verbal consent to protect their confidentiality. The research team contacted

participants via phone or email to schedule qualitative interviews. Participants received a $10 honorarium for study enrolment, $5 for the call to schedule an interview, and $30 for the interview. Thirty-three participants agreed to be contacted for an interview and 17 completed an interview. For the 16 other people who agreed to an interview, the team was not able to reach and/or schedule them for an interview. Participants' housing and financial precarity may have limited their ability to stay connected to programs and services. With this in mind, we shortened the follow-up period from 6 months to 3 or less months to increase interview completion, but we still could not reach the other 16 participants by telephone or email to make the final interview arrangements. The average time between program enrolment and interview was approximately 6.3 months (range of 1 to 21 months).

We contacted the two GACWs via telephone to request interviews. In March and April 2019, both GACWs provided informed written consent and participated in semi-structured interviews. GACWs did not receive an honorarium for their participation.

Client and GACW interviews were audio-recorded following informed consent. All study data (participant interviews and intake data) received a study identification number. A third party service transcribed the interviews verbatim.

## Data collection and instruments

During the program intake process, GACWs recorded client socio-demographic information in a study database. They also asked clients about their health, gambling activities, service use, and housing. GSM transferred these data to the research team at the end of data collection. Two research coordinators (male)(author GT, BA Criminology and DK, MA Sociology) conducted in-person qualitative interviews with GACWs and clients in case management at either GSM or St. Michael's Hospital between November 14, 2018 and April 24, 2019. The interviewers were trained in qualitative interviewing techniques (e.g., active listening, probing), research ethics, how to exercise sensitivity in working with populations with complex health and social needs. Interviews with clients were conducted between November 2018 and December 2019. GACW interviews were approximately 90 minutes in length and conducted in March and April 2019. Client interviews ranged from 17 to 60 minutes in length. Two semi-structured interview guides, one for clients and one for GACWs, included questions on the program as a whole (e.g., successes, challenges, and suggestions for improvement), specific features of the program (e.g., case management, CBT, LS), and the effectiveness of the program in meeting client needs.

## Data analysis

Descriptive statistics were calculated using means, standard deviations, proportions, and counts. Data is suppressed to ensure confidentiality when any cell count on a particular variable is less than 6.

We performed a qualitative content analysis of the interviews with program clients (n = 17) and GACWs (n = 2) to assess how they experienced and responded to the program. Qualitative content analysis involves interpreting "the content of text data through the systematic classification process of coding and identifying themes or patterns," and comprises three approaches: conventional (or inductive), directed (or deductive), and summative (or counting and comparing in context) [32: 1278]. The three approaches share roots in the naturalistic, interpretive paradigm, but one of the important ways they differ is in their analytic aims, and thus the selection of any particular approach depends on the research objectives. In our case, we were interested in participants' experiences of the GAP. Given the novelty of the program, the unique environment in which it was developed and implemented, as well as the limited

research on PG among people experiencing homelessness and housing instability, we adopted an inductive strategy to describe a phenomenon for which theory and research is limited.

Four members (AB, GT, PD, SHW) of the research team coded the 17 client interviews, and one member (AM) coded the 2 GACW interviews. The initial coding process identified both line-by-line codes and high-level themes within the data. Next, more focused coding—e.g., collapsing, expanding, and specifying initial codes—clarified and refined the respective coding frameworks for GAP client interviews and GACW interviews. Next we conducted detailed analysis of these coded data to identify, define, and illustrate themes, patterns, and connections within and across codes. Three coders (GT; SHW; PD) focused on the themes that addressed participant experiences in the program and one coder (AM) focused on the themes that addressed GACW experiences in program development and facilitation. Nvivo 10 was used to organize the data.

We identified two overarching themes that would inform the structure of the analysis that follows: program delivery and program impact. Program delivery included participant perspectives on the primary components of the program: individual case management and group sessions. Program impact included the influence of program participation on gambling behaviour, finances, and housing. Prior to detailing how the GAP program was received by participants and GACWs in the findings, we describe the study participants.

## Findings

### Description of study participants

We interviewed 17 GAP clients, the majority of whom were men (< 6 women). Participant age ranged from 37 to 80 years, and the mean age was 54 years. All participants were either experiencing homelessness (n = 11) or at-risk of homelessness (n = 6). The Canadian Definition of Homelessness includes two definitions of "at risk of homelessness":

- Precariously Housed: facing serious housing problems, including unaffordable housing, bad housing conditions, overcrowding, or unsafe housing; and

- At Imminent Risk of Homelessness: facing immediate potential loss of housing due to eviction, inability to pay rent, or violence in the home.

Most participants (n = 10) had completed some form of post-secondary education and were unemployed (n = 10), with their primary source of income from social assistance, known as Ontario Works or the Ontario Disability Support Program. When looking at the larger sample (n = 35) who participated in the study and program, less than half (n = 15) had completed some form of postsecondary education. Participants with postsecondary education are over-represented among those who completed an interview, relative to those who did not. The average monthly income was approximately $1000. Participants reported debts ranging from $2,000 to $80,000.

Health challenges were common. Over 70% of the participants reported physical health conditions, with multiple people reporting traumatic brain injury (n = 7), joint and/or back pain (n = 7), and hypertension (n = 6). Mental health and substance use challenges were also common, with 11 participants reporting a history of problematic substance use, and the majority (> 60%) reporting mental health issues. The most prevalent mental health issues included depression (n = 11), post-traumatic stress disorder (n = 8), and generalized anxiety disorder (n = 7).

Scores on the PGSI ranged from 6 to 25, with a mean of 15.4 (STD: 5.8). For the majority of participants (> 60%), the GAP was their first involvement in a gambling recovery program.

**Table 1. GAP components and levels of participation.**

| | Case Management | Life Skills | Cognitive Behavioural Therapy |
|---|---|---|---|
| Purpose and Techniques | Identify client-specific needs, strategies, and services. | Enhance participant skills to manage everyday challenges, including the management of gambling activities. | Introduce general CBT principles and apply CBT techniques in the context of persons experiencing PG, poverty and homelessness. |
| Topics Covered | Identifying gambling problems, causes, and consequences; developing personalized pathways for recovery; and considering additional participant needs (e.g., employment, finances, housing, and health). | Understanding gambling and its impacts; anxiety and triggers, mindfulness, communicating with others; asking for help; thinking ahead to anticipate self-needs; managing conflict; engaging in self-care; debt counselling; budgeting; cooking classes; and healthy aging. | Defining and identifying problem gambling; gambling as learned, habitual behavior; identifying gambling triggers and high-risk situations; employing self-management strategies; correcting gambling-specific thinking errors; solving problems and setting goals; managing negative emotions; and dealing with relapse. |
| Delivery Method | • Individual<br>• 1-hour meeting between GACW and client | • Group<br>• 1.5-hour meeting<br>• GACW led | • Group<br>• 1.5-hour meeting<br>• GACW led |
| Number of Sessions Offered | • 1 session per week between GACW and client | • 1 session per week over 7 or 8 consecutive weeks | • 1 session per week over 7 or 8 consecutive weeks |
| Session participation[1] | • 16 participants attended case management<br>• 2 to 15 hours of case management was completed, averaging 6 hours across clients | • 12 participants started and 7 completed life skills group work out of 17<br>• 1 to 6 Life Skills group sessions were completed, averaging 3.5 sessions across clients | • 12 participants started and 6 completed CBT group work<br>• 1 to 8 CBT group sessions were completed, averaging 5 sessions across clients |

Participants engaged in multiple types of gambling, but the most common included lottery (n = 12), instant win/scratch tickets (n = 11), slot machines (n = 11), casino table games (n = 10), sports betting (n = 9), and horse or dog racing (n = 6).

## Program components

Program components included group work on shared challenges around gambling and case management for individualized supports. Table 1 presents an overview of the therapeutic components including details on delivery method and number of sessions offered for each and their respective levels of participation among study participants.

**Case management.** Both GACWs and program participants stressed that mutual trust and commitment were required to realize the full potential of one-on-one counseling and effect significant change in gambling behaviour. Program participants emphasized GACWs cultivated trust through their personal and professional qualities which gave case management sessions "a talk with a good friend" quality and created a space where participants could share experiences that they would not otherwise disclose. For example, one participant said that the GACW expressed a genuine confidence in and concern for her and that created a comfortable space to discuss problems, issues, and potential solutions in an open, honest, and accountable yet amicable way:

> *All my life I have a hard time to believe in people and [my GACW] always believes me . . . We discuss a lot of stuff . . . like what you do for today, how my situation might explain that it's [gambling] still a big problem [and] it's affect [on] my life, but [my GACW] never push me, like really push me, like you need to sit down—no, not like that [Laugh]. [Our sessions are] more like for my opinion, talk[ing] with [a] good friend.* (304)

Similarly, another participant felt that the listening abilities and relational qualities of the GACW fostered an environment that encouraged greater levels of disclosure in case management sessions:

*I've shared stuff with [my GACW] that I've probably never shared with anybody. . . [my GACW] made me feel comfortable, like [my GACW] is listening.* (123)

Some program participants recounted experiences of poverty and/or homelessness related marginalization in gambling programs outside of GSM that focused on PG as a specific problem-condition to the neglect of other issues, such as homelessness, poverty, mental health, substance use, and disability. GACWs and the shelter-based GAP, in contrast, emphasized treatment of the "whole person":

*. . . one of the things that the clients mentioned to us early on . . . was that, you know, this individual had been to other programs, but bigger organizations for gambling treatment, and felt treated better by us . . . because of the stigma of being homeless . . . I think listening to people who have previously been ignored or treated badly is a main factor in why they keep coming back to us . . .. almost all of my clients . . . have more than one issue. I mean, not only are they all in poverty and they all have gambling issues, but some of them have mental health issues . . . a lot of them have family conflict, a lot of them are unemployed or have a disability of some kind. So to just treat the gambling would be to not treat the whole person.* (CW 1)

Still, while program participants felt valued and understood in case management sessions, accountability was also an important part of the treatment relationship and the rhythm of recovery, for the regular meetings provided structured opportunities for re-motivating, -focusing, and -directing participants. For example, one participant said that the regular meetings with the GACW were helpful in staying on track:

*When somebody's out there trying to motivate you and help you in ways that you cannot waste your time but take time. I found that [my GACW] is extremely, extremely helpful one-on-one.* (115)

One way GACWs facilitated participant commitment to the harm reduction and recovery process was through regular goal-setting:

*There was some pretty deep discussions . . . so I'm pretty thankful for [those]. And then mostly about thinking about things properly, and thinking about things in order . . . in terms of my goals . . . what I can actually start doing with [my] money if I'm not gambling. . . but not to reach out too far, like just stay in reality, you know, like 'right now' . . . What about right now?* (113)

Focusing on specific, practical and realistic recovery goals and strategies in the context of poverty and/or homelessness informed everyday decision-making and grounded aspirations of a future beyond PG.

Aligned with GSM principles of client-centered care, GACWs adapted their approaches to care to meet the individual needs of clients:

*My client . . . right now is working. . . . He said to me that it was very important the support he had [in case management] . . . because of that I opened up the option of doing phone calls, sessions like coaching . . . it worked very well for him and I think that is also important for those who cannot make it [to in-person case management].* (CW 2)

**Group meetings.** CBT and LS group meetings were key GAP components. Compared with their experiences at other organizations, GACWs noted that GAP participants seemed to be particularly engaged in the group meetings:

*Clients really like the groups. [They] are willing participants . . . [and] give us good feedback . . . [For example,] the most recent Life Skills group was rated . . . 94% approval.* (CW 1)

Participants found the content and exercises relevant to their everyday lives. For example, a participant described learning and incorporating CBT strategies to manage and modify undesirable habits:

*they talked about the 21 days and how you can re-program yourself in 21 days to change a habit . . . the result of that is I do 21-day affirmations right now and stuff like that too.* (113)

Another participant noted the everyday utility of the LS group materials:

*So all those [Life Skills] topics were amazing. The, especially the budgeting like the accounting, the budgeting, the, you know, just simple life skills, like eating healthy, like shopping and, you know, looking for, you know, the sales.* (108)

Similarly, participants felt that groups were collegial, collaborative, and co-productive, promoting cross-fertilization of perspectives. That is, sharing with and learning from others experiencing PG and precarious housing provided personal opportunities to reflect and work on one's own issues:

*What worked for me is . . . having different people in the group. . . if you keep an open mind you can see that how it triggers different people at different surroundings and you're able to, you know, like not crosstalk, but you can talk . . . it's helpful because you go home with something . . . something you could dissect and say, 'Hey, I'm in a better place today by not gambling or not picking up,' you know?* (109)

*[The Life Skills group helped by] being able to hear some of the people giving you advice, giving you solutions, feeling connected with others, learning new skills, and feeling that somebody cares about helping you.* (124)

There was thus a rhythm to program participation as the group gathered to develop self-awareness, practical strategies, and emotional energy, dispersed to incorporate course knowledge into everyday practice, and returned to be accountable to self and other. Relatedly, one participant stated that the relational expectations as well as the structured schedule of the program experience increased the sense of personal responsibility:

*I want to be able to stop this . . . [I] enrolled myself . . . I've done it . . . [And] you know what? If you're coming to a class, and explaining myself, and like showing up on time, [it] gives me responsibility.* (123)

Although the LS and CBT group content and activities were generally well-received, GACWs were receptive to participant feedback, attempting to align programming with participants' needs and interests. GACWs also modified meeting schedules to promote greater attendance and participation (e.g., minimizing scheduling conflicts with common "cheque days" for social assistance payments). Some clients cited that the adaptive approach to content and

care accounted for their participation in subsequent cycles of LS and CBT groups. For example, one participant said that the GACWs responded to participant feedback recommending revisions to the program content and that helped to keep the group meetings fresh and engaging:

*They [GACWs] listened and they took advice, and they changed a few things in the program . . . so each time you go it's a little bit different, so that's why I come also is that things change and they change up and they make things a little more interesting. It's not like a script—the script changes. There is no script. It's day-by-day like stuff, you know what I mean? (301)*

Similarly, another participant found that the revisions to the Life Skills material and delivery warranted taking the course twice:

*It's my second round of that [Life Skills group] and I found out it's getting more interesting somehow—how they prepare, how they prepare the topic, and from that topic how they present information. (304)*

## Program outcomes

Focusing on three central program outcome categories—gambling, finances, and housing—we consider below some of the ways the GAP became an important resource for participants in confronting and managing the challenges of PG in the context of poverty and homelessness.

**Gambling matters: Increasing gambling awareness and reducing gambling behaviour.** Through the GAP, participants developed a better understanding of their gambling behaviour and its attendant connections to other areas of their lives. Participants, for example, became more aware of the interrelatedness of their gambling, financial, and housing problems:

*[I've learned] that [gambling] was destructive in my life, that it was actually causing me [other] issues . . . I didn't believe that the scratch ticket thing was a problem until recently. Right, like spending X amount of dollars and then adding it up, it ends up being like over one-hundred, two-hundred bucks a month, which if I had not been gambling, I would have been able to pay my rent. (120)*

Participants began to learn and practice a variety of strategies for self-managing problem gambling—that is, learning how to manage PG symptoms, interventions, consequences in everyday life, as an adjunct or alternative to formalized treatment. For example, consistent with cognitive therapy approaches to problem gambling, one participant noted the program introduced cognitive strategies to identify and disrupt patterns of thought characteristic of PG:

*It [the GAP] helps you, like in your thinking process . . . it points out in the program . . . how your thinking process works when you're a gambler and how to try and stop it from happening and . . . sometimes now I'm successful at doing it . . . I start thinking, you know, I've been wrong so many times and this one's probably not going to happen either. . . (300)*

Other strategies included identifying and avoiding triggers, keeping busy, and recognizing and managing urges. For example, one participant noted the effectiveness of keeping busy in reducing urges:

*If I keep myself busy, I don't have the urge to gamble. But this month I haven't actually had the urge to gamble and same as last [month]. I got paid on the end of March, so from March*

*28 I haven't been, I haven't got the urge to gamble as much as I used to since I've been coming to this group so it's actually been helping me.* (123)

Similarly, another participant was better able to resist urges by delaying gambling action and instead reflecting on the best course of action given the likely outcome and consequences of gambling:

*I learned that if you, if I feel like going gambling, I'll just sit down and say 'Okay, wait a minute. Let's slow down.' And like it works, but you know I think about it. I never used to think about gambling. It was all like when you lose all your money you feel like shit, blah, blah, blah. I used to think about the winning, but you have to, you're not going to win every time. You're going to lose most of the time.* (121)

**Financial matters: Reorienting relationships with money.** Participants also cited improvements to their financial circumstances as they acquired new perspectives on gambling and money, developed new strategies for limiting the financial harms of PG, and connected to financial specialists through the GAP. The GACWs supported participants in reorienting their relationship with money through case management and group sessions:

*I started [to treat my money more responsibly] ever since I am with this [GACW] discussion. I started having that understanding that I got to be responsible about my finances . . . I at least save myself from looking at gambling as a source of income, number one. That's the biggest, biggest achievement I've already achieved by associating with these people.* (117)

Some participants noted that practicing transparency and accountability in their interactions with GACWs could support the emerging improvements to their financial situations. For example, one participant described marked improvements to personal finances as a result of the regular discussions with the GACW about weekly savings versus gambling expenditures:

*[My financial situation has improved in terms of] the amount of money that I've been able to save and be accountable as I show my case manager. I'm accountable to, you know, somebody that could be like, 'Hey, what the hell are you doing? What's going on there?'* (108)

GACWs worked with participants to develop tailored strategies to reduce the risk of financial harm. For example, following a GACW suggestion in case management, one participant adopted an approach of strategic inaccessibility to protect personal belongings that would otherwise be likely candidates for liquidation amid gambling urges:

*[Case management] kept me from selling personal items to get money to gamble. So, yep, not that I have a lot of personal items right now or that aren't at my parents' house, which is up in [a rural area], so I don't have access to them. But when I did have access to them it encouraged me to put them there, so I couldn't sell them and get rid of them to gamble . . . and that was also partly [a GACW] suggestion, like 'Keep your valuables [somewhere difficult to access while gambling]. Why would you sell them to risk losing money?'* (120)

When participants integrated these strategies into their repertoire of everyday perspectives, practices, and approaches to problem solving, they realized additional dividends beyond,

though still related to, an improved financial situation, such as fewer gambling urges, and greater savings for basic needs:

> *I've actually got money in the bank for the first time. My cheque that I got—I still had money left over, and I haven't gambled any away . . . [My strategy has been to] try saving money and try avoiding those places [in which I have tended to gamble]. That's what I've been doing, and I've actually, I still have a couple hundred dollars in the bank from $340. I've still got $200 in the bank, so I haven't really had the urge to gamble. I didn't even have the urge to gamble today.* (123)

> *I always got money in my pocket now. It's not going to gambling . . . my fridge is full now.* (100)

Notwithstanding these improvements to participants' financial situations, fixed, limited incomes presented enduring everyday challenges. Most participants were dependent on government financial supports that did not cover the basic cost of living in the city:

> *I mean the need for more money is one of the biggest things for my clients because a lot of them are, they're almost all on fixed incomes and those incomes are just not enough, to keep up with the cost of living . . . [They] have no choice but to live in a shelter because their total monthly income is less than the average rent in Toronto, so there's no possible way for them to pay for housing and food and their other needs on the fixed incomes that they have so that's one of the biggest ones. . ., their income.* (CW 1)

With such tight financial constraints, there was often little to no margin for error in financial decision-making without risking serious personal consequences, such as losing housing or going hungry. Case workers felt more comfortable and qualified in giving general, rather than specific, financial counseling, referring clients who needed more involved and tailored financial counsel to specialists who could assist with resolving debts, claiming bankruptcy, and holding money in trust. However, low client interest and a long trustee waitlist limited the potential impact of the volunteer trusteeship associated with the GAP:

> *I've brought it [volunteer trusteeship] up with several of [my clients] . . . Some of them have expressed interest, but . . . none of them have followed through . . . The other issue is that our trustee at Good Shepherd has a really long waitlist . . . although I think the biggest barrier for our, my clients specifically, was just [many were] not interested in it . . . [For example] some of them have said . . . 'I don't want to lose control of my money' . . . When they say that, I ask them 'Okay, so tell me how much control [do] you have over your money when you're gambling' . . . and the answer is 'Well, obviously I don't have control over my money when I'm gambling, but I feel like I do because it's in my pocket when I start gambling' . . . as opposed to in the trustee's account . . . So, they feel like they have more control even though at the end of the day they've lost it all.* (CW 1)

Still, from the GACW perspective, while the volunteer trusteeship initiative is underutilized resource, it has the potential to be an effective tool for mitigating PG. For example, when participants pursued specialist referrals, they tended to realize tangible improvements to their financial situations:

> *She [financial counselor] did help me set up a repayment plan with the government. I owe the government $55,000 and now I only have to pay them $1,600, so I liked that part about it.* (108)

**Housing matters: Seeking, securing, and stabilizing shelter.** Housing was a priority issue. Both GACWs and program participants recognized that housing serves as a platform from which people can pursue personal goals and improve the quality of their lives. For participants who were just a few weeks into the program, many were experiencing homelessness and working toward securing housing with support from the GAP and related supports at the shelter:

> *It's not stabilized yet, but we're working on that process . . . we're headed in the right direction because I'm happy where we're at right now.* (109)

Some participants of the GAP secured and maintained housing while working their way through the program. Housing provided them with the stability needed to envision and strive toward a life beyond PG, poverty, and homelessness. As one participant described, the GAP and related shelter services gave him a "safe" place to "make that next step into my life" (108). Another participant said his life situation changed "180 degrees" (121) after securing housing. Stable housing provided sure footing for setting goals, managing time, avoiding drugs and alcohol, and seeking employment:

> *Before, I was just living in [a shelter] wondering if I have a room if I didn't make it back here by 5:30 [registration for beds], how to answer questions about what I'm doing, where I'm going. Whereas now [living in "social assisted housing"] it's like I got to set goals. I just get up. I work part-time. [I'm] looking for full-time, but my current few hours are my own judgment. I get home when I want to get home, I leave when I want to leave. It's like, it's a better situation for me and as far as the drugs and the alcohol and like the cards and the gambling.* (100)

> *Okay, so six months, seven months ago when I started in the program, I was homeless, I had nobody, I lost my job, all of the information, like I lost everything, [including] my sanity . . . I lost it all. Now, seven months later . . . I've been able to save money, buy a car, get back to work, get a condo with a mortgage, pay back the government the money that I owed, paid back family the money that I owed just by working, [by attending] the program and not gambling.* (108)

From the GACW perspective, stable housing was the most difficult need to address, for there are many barriers to housing in the population (e.g., poor credit, low income, limited-to-no savings for first and last months' rent, difficulty saving, limited availability of affordable housing, poor quality low-income housing stock). GACWs felt there was little they could do to ameliorate the precarious housing situation of most participants. Further, participants who secured housing tended to stop attending and contacting the GAP, and that made tracking program effectiveness regarding housing stability and eviction prevention difficult. Still, one case worker spoke of a client who was facing potential eviction because his disability support cheque was delayed during a postal strike. The GACW coached the client so that the landlord was paid and eviction was avoided.

GACWs cited examples like the one above to demonstrate the personal significance of these ostensibly small victories. Each step forward may represent meaningful change to participants' understanding of themselves and their recovery. That is, in confronting and managing gambling, financial, and housing matters, participants were also attending to self-matters, such as self-care, self-identity, and self-esteem. For example, the GAP helped one participant (re) achieve a more authentic sense of self, less encumbered by gambling problems:

*[The GAP] just helped me to be complete. It helped me to get housing. It helped me to just be able to enjoy and live life on life's terms. It helps me to stay focused. It helps me to stay motivated. I know that every week I have a meeting here that, you know, I can talk about whatever and anything . . . Most importantly, it's just helped me to become the real me again, you know, rather than being under the boot of gambling or addiction. It just allows me to take that boot off and just to be able to enjoy life and live life.* (108)

Similarly, another participant improved a sense of contentment and balance in everyday life:

*Just in life it's made me feel a more, a better, more balanced person. I don't have that discontent when you go and you don't win. You don't have that let down feeling. I find that I'm finding other things to do, like going for nice long walks or I'm volunteering a little bit. I'm finding more positive ways to spend my time.* (115)

## Discussion

The purpose of this article was to report on the results of an evaluation of an innovative pilot GAP that was situated within a multi-service agency within a shelter setting and designed to address PG within the context of poverty and homelessness. Until the implementation of the GAP at GSM, there were no dedicated PG treatment options for people experiencing poverty and homelessness. The novelty of the GAP underscores the need for researchers, practitioners, and policymakers to attend to, and learn from, the experiences of the program facilitators and participants. Drawing on qualitative interview data, the article presented GACW and GAP participant perspectives on the program components and outcomes.

The emphasis in case management was on a collaborative process of identifying needs, developing strategies, and connecting to services. Consistent with client-centered practices [33–35], GACWs adapted their approaches to suit the varying needs of clients. In doing so, GACWs attended to the interrelationship of PG and other complex needs, such as homelessness, poverty, mental health, substance use, and disability. Group meetings helped participants develop self-awareness by the act of listening to peers facing similar challenges. GACWs and program participants leveraged the collective experience of the group to illustrate important concepts and think through problems. In terms of program outcomes, the responses of GACWs and participants suggest that the GAP facilitated client awareness of gambling behaviours and harms. Participants also learned money management strategies to gain greater control of their financial well-being amid fixed and limited incomes. Relatedly, while housing remained a challenging issue to address, GACWs helped several clients avoid eviction or secure housing.

Our data suggest that PG treatment within the context of poverty and homelessness benefits from an approach and setting that meets the unique needs of this community. For example, the environment within GSM is distinctive in its coordination of care for multiple health and social issues, including chronic illness and disability, mental illness and addiction, finances, food and clothing, and emergency shelter and affordable housing. The integration of gambling treatment into this multi-service delivery model addressed the complex needs of the service users, reflecting a call among service agencies for integrated and person-centered approaches to care that respond to client needs, foster therapeutic relationships, reduce experiences of discrimination and stigma, and enhance recovery [1,36]. In developing the GAP, GSM drew, not only on evidence-based approaches to PG treatment, but also extensive experience working with the target population.

The GAP demonstrated early promise in supporting participants in the process of recovery. Addiction recovery is not linear, but rather involves cycles of harm reduction, abstinence, and relapse [37–39]. This evaluation identified important incremental measures of PG intervention success for people experiencing poverty and/or homelessness, including engaging with the program, increasing awareness of the harms of gambling, recognizing oneself as a "gambler," moving toward recovery, developing therapeutic relationships, establishing support networks, managing finances, and stabilizing housing. These shifts in gambling selves and situations are significant, especially given the hidden nature of gambling addiction and the challenges participants face in their everyday lives that contribute to low rates of treatment seeking in the population [40,41].

The complex realities of experiencing poverty, homelessness, mental illness, PG and substance dependence are essential to consider when designing and implementing evaluations of interventions like the GAP. Standard evaluation models are not sufficient when innovation is inherent within the intervention because these initiatives ". . .are often in a state of continuous development and adaptation, and they frequently unfold in a changing and unpredictable environment" [42]. Many people experiencing homelessness, poverty, and PG contend with health and social issues that challenge their ability to seek care and adhere to treatment programs. On any given day, they may lose their housing, experience a mental health crisis, and/or relapse in their gambling or substance use. Program design, development, implementation, and evaluation should attend to these challenges. A strength of this study is that the team worked with the community partner integrating their feedback into all aspects of the evaluation design, and implementation to ensure the tools were relevant, sensitive to the needs of the clients and adaptive to the unique circumstances. For example, as researchers we faced challenges to reconnect with some clients, even just weeks after their enrolment into the intervention, so we adapted by shortening the follow-up period. We also offered participants honorariums at study enrolment ($10), when we called to schedule an interview ($5), and at the end of the interview ($30) to support them to engage in the research. Such challenges will continue to emerge as the program develops to support new clients going forward. Future evaluations must account for the unpredictable, precarious contexts in which people live and adapt indicators of success accordingly [42].

We also acknowledge the limitations of this evaluation, particularly in the context of the challenges we noted above. First, the GACW commitment to client-centred support involved continual assessments of and adjustments to program content and delivery based on clients' evolving needs and responses as is reflective of a client-centred approach to care. Given the dynamic nature of clients' lives, the GAP delivery and content continued to change in subtle ways throughout the duration of the study, giving our object of inquiry an emergent quality that made it more difficult to pin down in precise and encompassing ways. Second, and as mentioned earlier with respect to maintaining participant contact and scheduling interviews, the study of PG among populations facing manifold personal, social, and health challenges often poses additional recruitment and data collection challenges than researchers might encounter in general population or treatment samples. We experienced and adjusted to these challenges throughout the study, but there were 16 people who we could not reach for interviews, which may have affected the range of perspectives we could solicit in performing this evaluation. Third, one of the contributions of the evaluation is tied to the novelty of the GAP, not only in terms of its focus on PG in the context of poverty and homelessness, but also in terms of where the program was situated: a multi-service agency within a shelter setting. While there is much to learn from innovative programs, we also acknowledge the limitations of focusing on a single research site, especially one serving primarily men.

The burden of harm from problem gambling weighs heavily on those experiencing poverty and homelessness. GSM uncovered this hidden issue among their clients and sought to close an identified gap in services to treat problem gambling. In the absence of PG programs tailored to the specialized needs of the population they serve, they developed and implemented the GAP, a first-of-its-kind program that offers a model for potential wide-scale roll out. On a system-level, however, the current state of PG services reflects an enduring failure to recognize PG as a public health issue requiring greater awareness, coordinated screening, and sustainable funding for innovative models of care that integrate services and address the needs of special populations. As an important and often hidden public health concern [13,43,44], it is imperative to implement PG screening into clinical and social services. Cross-sector collaboration is needed to facilitate improved integration and coordination across service sectors (e.g., family support, immigration and drug and alcohol services) for those experiencing PG, comorbid conditions, and poverty [45].

Services tend to develop in organizational, institutional, sectorial silos, resulting in a lack of communication, coordination, and collaboration in developing and delivering models of care. Bridging service silos may offer opportunities for enhanced PG awareness, care, and outcomes. Integrated approaches to care are particularly important to address challenges of special populations, supporting the physical, psychological, and social needs of persons as interrelated determinants of personal and public health. The results of programs that bridge service silos and provide community-oriented approaches to care have been promising. One study shared four features that contributed to the success of five programs in achieving the "Triple Aim" of better care, improved health, and lower costs: shared leadership, shared data, shared commitment to person-centered care, and flexible financing [46,47].

In the context of the GAP at GSM, for example, partnerships with agencies serving similar populations, especially proximate agencies within walking distance, may provide not only access to a larger pool of potential clients but also opportunities for enhancing problem awareness, sharing program leadership, coordinating screening, sharing program costs, and increasing program availability. A coordinated approach is especially important given that the complex and diverse challenges with which people experiencing PG, poverty, and homelessness contend often make it difficult to provide population-wide support through any single agency.

System-level changes to PG program funding, development, and coordination may be slow to arrive. Alternative approaches may be adopted to bridge the gap between the current lack of services to future coordinated system-level responses. For agencies that currently do not provide gambling addiction services, GA may offer an opportunity to meet some of the needs of these clients without the added costs of program development and delivery.

## Conclusion

This evaluation illuminated the complex nature of the lives of people experiencing problem gambling, homelessness and poverty. Within a short timeframe, the GAP supported participants in the process of recovery, enhancing their understanding and control of their gambling selves, behaviours, and harms. This project demonstrates that gambling within the context of poverty requires a unique treatment space and approach. GSM has developed a service model that meets the needs of people experiencing poverty/homelessness and problem gambling. The integration of gambling into a multi-service delivery model enriches the suite of services offered by GSM. The GAP stands as an effective, comprehensive approach to care and a model to emulate within other shelter service agencies.

## Supporting information

**S1 File.**
(PDF)

**S2 File.**
(PDF)

## Acknowledgments

The authors gratefully thank the clients and case workers for sharing their experiences. We also grateful for the collaboration of Good Shepherd Ministries on this project. The project was supported by the MAP Centre for Urban Health Solutions, St. Michael's Hospital and the Dalla Lana School of Public Health and Centre of Criminology and Sociolegal Studies, University of Toronto.

## Author Contributions

**Conceptualization:** Flora I. Matheson, Sarah Hamilton-Wright, Tara Hahmann, Arthur McLuhan, Aklilu Wendaferew.

**Data curation:** Flora I. Matheson, Sarah Hamilton-Wright, Guido Tacchini, Aklilu Wendaferew.

**Formal analysis:** Flora I. Matheson, Sarah Hamilton-Wright, Arthur McLuhan, Guido Tacchini, Parisa Dastoori.

**Funding acquisition:** Flora I. Matheson, Sarah Hamilton-Wright, Tara Hahmann, Aklilu Wendaferew.

**Investigation:** Flora I. Matheson.

**Methodology:** Flora I. Matheson, Sarah Hamilton-Wright, Tara Hahmann, Aklilu Wendaferew.

**Project administration:** Flora I. Matheson, Sarah Hamilton-Wright, Guido Tacchini.

**Resources:** Aklilu Wendaferew.

**Software:** Sarah Hamilton-Wright, Parisa Dastoori.

**Supervision:** Flora I. Matheson.

**Visualization:** Tara Hahmann, Arthur McLuhan.

**Writing – original draft:** Flora I. Matheson, Sarah Hamilton-Wright, Tara Hahmann, Arthur McLuhan.

**Writing – review & editing:** Flora I. Matheson, Sarah Hamilton-Wright, Tara Hahmann, Arthur McLuhan, Guido Tacchini, Aklilu Wendaferew, Parisa Dastoori.

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
