## [Decision Letter · Decision Letter 0]

24 Sep 2021

PONE-D-21-07198Filling the GAP: Integrating the Gambling Addiction Program into a shelter setting for people experiencing poverty and homelessnessPLOS ONE

Dear Dr. Matheson,

Thank you for submitting your manuscript to PLOS ONE. After careful consideration, we feel that it has merit but does not fully meet PLOS ONE’s publication criteria as it currently stands. Therefore, we invite you to submit a revised version of the manuscript that addresses the points raised during the review process.

The manuscript has been evaluated by three reviewers, and their comments are available below.

The reviewers have provided some comments that need attention. They request additional information about definitions used in the study, and details of the study setting and participants. They also request some amendments to the quality of the reporting of the Results.

Could you please revise the manuscript to carefully address the concerns raised?

We look forward to receiving your revised manuscript.

Kind regards,

Marianne Clemence

Associate Editor

PLOS ONE

Journal Requirements:

3. When reporting the results of qualitative research, we suggest consulting the COREQ guidelines: http://intqhc.oxfordjournals.org/content/19/6/349. In this case, please consider including more information on the number of interviewers, their training and characteristics; and please provide the interview guide used.

Furthermore, please provide additional details regarding the participant eligibility criteria.

4. Please provide additional details regarding participant consent. In the ethics statement in the Methods and online submission information, since informed consent was verbal/oral, please specify: 1) whether the ethics committee approved the verbal/oral consent procedure, 2) why written consent could not be obtained, and 3) how verbal/oral consent was recorded. If your study included minors, please state whether you obtained consent from parents or guardians in these cases. If the need for consent was waived by the ethics committee, please include this information.

5. Please ensure that you include a title page within your main document. You should list all authors and all affiliations as per our author instructions and clearly indicate the corresponding author.

Reviewers' comments:

Reviewer's Responses to Questions

**Comments to the Author**

1. Is the manuscript technically sound, and do the data support the conclusions?

Reviewer #1: Yes

Reviewer #2: Yes

Reviewer #3: Yes

2. Has the statistical analysis been performed appropriately and rigorously? 

Reviewer #1: N/A

Reviewer #2: No

Reviewer #3: N/A

3. Have the authors made all data underlying the findings in their manuscript fully available?

Reviewer #1: No

Reviewer #2: Yes

Reviewer #3: No

4. Is the manuscript presented in an intelligible fashion and written in standard English?

Reviewer #1: Yes

Reviewer #2: Yes

Reviewer #3: Yes

5. Review Comments to the Author

Reviewer #1: The manuscript entitled ‘Filling the GAP: Integrating the Gambling Addiction Program into a shelter setting for people experiencing poverty and homelessness’ is a clearly written appraisal of a programme intervention. The rationale for the need to create this intervention programme and details of the programme are explicit; the design and method are clear and sample recruitment is also clearly stated. The programme intervention is a complex one due to the nature of the presenting clientele. The authors acknowledge the intricate systemic web of homelessness, disability, unemployment, and problem gambling and do so with due regard for the dignity of the sample. It is interesting to note that close to two-thirds of the interviewees had post-secondary qualifications. Given the small sample, not much else can be said but it is an interesting bit of information and gambling addiction case workers can tailor their interventions accordingly. Verbatim extracts showcase the verbal ability of some of the clients and given the general level of education among the sample, perhaps the programme can be tailored even more. The iterative and adaptive nature of the programme is to be commended and although it is acknowledged as a potential limitation, it can equally be construed as a necessary adaptation of the programme to ever-changing needs of the clientele. It seems, from the verbatim text, that some clientele found this to be a positive aspect of the programme and further underscored the need to take on board what the clients are saying. The incentive structure is noteworthy where participants received a $10 honorarium for study enrolment, $5 for the call to schedule an interview, and $30 for the interview. The assumption is that participants who fell out of the study part of the way were nonetheless compensated for their time. As soon as housing is secured, attrition rates increase. The latter issue in tandem with the incentive structure could perhaps be looked at a little more skeptically. The range of time spent in the programme is quite large with some participants clearly not involved for any meaningful length of time and others likely benefitting handsomely. The authors do acknowledge that the programme purposely tried to avoid scheduling conflicts (clients receiving benefit payments on days which would have precluded their involvement in the programme). Clearly, the clientele is diverse and given the varied needs, the programme is likely to be more useful to some than others. An interesting finding is the lack of engagement with the financial trustee: “However, low client interest and a long trustee waitlist limited the potential impact of the volunteer trusteeship associated with the GAP”. This is certainly an avenue worth pursuing in terms of increasing engagement with trustees. It is possible that the recent global financial crisis and the reputation of banks has played a role. The fact that participants disengaged from the programme after housing was secured is of concern. Was it felt that programme engagement was perhaps mandatory to securing housing? It may be the case that participants may have felt that engaging with the programme would have increased their chances of securing housing. The authors acknowledge that “addiction recovery is not linear, but rather involves cycles of harm reduction, abstinence, and relapse” and this clearly makes programmes of this nature very complex. Another noteworthy conclusion of the study was the recommendation to improve inter-agency communication and leadership and very importantly to make available shared data – the last aspect is very difficult to address. The complex nature of the issue (homelessness, problem gambling, comorbidity etc) requires a multipronged approach. Sadly, basic things like housing requires political will. Many of the prevalent issues mentioned are the effects of multiple causes from the top down. Nonetheless, programmes of this nature do at least try to grapple with the effects. The research team are to be applauded for their efforts. The assumption with this manuscript is that this is a trial-run of sorts. More data is needed to truly evidence sustained impact.

Reviewer #2: This paper provides further insight into a meaningful and oft-overlooked area of gambling—problem gambling among those who are homeless or at-risk of becoming homeless. In addition, the paper provides a rich description of the incorporation of a gambling recovery program into a robust multi-service agency. The authors demonstrate that it is possible to successfully incorporate a gambling treatment program into such an environment and that it addresses a significant recovery component for many homeless people. They also provide nuance in describing the challenges of implementing such a program in a real-world clinical setting. The authors do well in describing their process and outcomes and the qualitative approach provides further dimension into the experiences of those struggling with homelessness, PG, and other comorbidities.

However, the following issues need to be addressed before further consideration for publication can be made.

First, please provide information about how many possible participants declined to answer the NODS CLiP. Likewise, the authors state that 33 participants consented to the interview. How many declined?

Second, please provide the actual number of men participating in the study. The authors only state that the majority were male and that overall 90% of the GSM are male.

Third, please provide an definition for at-risk of homelessness.

Fourth, the authors state that various descriptive statistics were used, but few were reported. It would be informative to have an idea of PG severity among this sample. A count of the number of participants scoring 1, 2, or 3 on the NODS-CLiP and the mean and SD of the PGSI would provide further clinical information of the sample. Please provide this information in the "Description of Study Participants" section.

Reviewer #3: Thank you for the opportunity to review this study. This is a very important and understudied topic and I appreciate the care the research team takes in identifying the issues faced by those facing housing instability and poverty. The paper is carefully and clearly written. I have only minor suggestion that may improve the paper.

First, I think it would be a good idea to define exactly what you mean by “problem gambling”. Making sure that a clear definition is given so that it does not get conflated with Gambling Disorder or pathological gambling would be a helpful primer for the reader.

Second, I wondered if any consideration was given to the data collection period. I know that challenges for those facing instability in their housing options vary from season to season in a place like Toronto.

The level of some post-secondary education seemed quite high in the client sample. I would be interested to see how this relates to existing literature/estimates for homeless populations and whether it might impact study/program participation.

Finally, in several cases the relevance of the excerpts to the themes being discussed were presented as self-evident. It would be helpful to the reader for the writing team to make a clearer connection between the theme and the evidence of that theme. One example is the excerpt at the top of page 15 (case 304). An explanation of the variability of content and its importance to keeping this client engaged would make the connection of evidence to theme more explicit and do more to demonstrate the value of a tailored approach.

Minor style points

Repetition of integration in abstract “The integration of gambling treatment into this multiservice

delivery model addressed the complex needs of the service users through integrated and

person-centered approaches”

Progrom: change to program.

Pg 3: well positioned= well-positioned

The acronym GSM is used before it is defined

Page 5: “GACWs performed outreach at other community agencies serving the target

population, which provided the basis for external referrals to the program.” Change “,which” to “that”. The use of which instead of that happens a lot throughout the manuscript

6. PLOS authors have the option to publish the peer review history of their article (what does this mean?). If published, this will include your full peer review and any attached files.

Reviewer #1: **Yes: **Raegan Murphy

Reviewer #2: **Yes: **Steven D. Shirk

Reviewer #3: No

---

## [Author Response · Author response to Decision Letter 0]

6 Jan 2022

Below we provide detailed responses to the reviewers’ comments. We appreciate the details provided by the editor and reviewers and feel these have greatly strengthened the paper. to the editorial comments and the reviewers comments we provide the following: 

• Journal Requirements:

A. Editor comments:

Response: We reviewed and applied the style templates.

Response: This has been completed.

3. When reporting the results of qualitative research, we suggest consulting the COREQ guidelines: http://intqhc.oxfordjournals.org/content/19/6/349. In this case, please consider including more information on the number of interviewers, their training and characteristics; and please provide the interview guide used. Furthermore, please provide additional details regarding the participant eligibility criteria.

Response: Additional information on the number of interviewers, their training and characteristics and on participant eligibility has been added in the recruitment section (lines 163-169). 

4. Please provide additional details regarding participant consent. In the ethics statement in the Methods and online submission information, since informed consent was verbal/oral, please specify: 1) whether the ethics committee approved the verbal/oral consent procedure, 2) why written consent could not be obtained, and 3) how verbal/oral consent was recorded. If your study included minors, please state whether you obtained consent from parents or guardians in these cases. If the need for consent was waived by the ethics committee, please include this information.

Response: In our research with people experiencing stigmatized identities, we often use verbal rather than written consent to offer an added level of protection for participants. We revised the information in the manuscript as noted below. Please see lines 96-101. We also provided similar information in the online system.

“We obtained verbal, rather than written consent, to offer an added layer of protection for participants who experience stigmatized identities. The project, including the verbal consent procedure, was approved by the Research Ethics Board [REB # 17-058] of St. Michael’s Hospital, Toronto, Ontario, Canada. Verbal consent was recorded in the consent form and on the recruitment database.”

All participants were aged 18+

5. Please ensure that you include a title page within your main document. You should list all authors and all affiliations as per our author instructions and clearly indicate the corresponding author.

Response: This has been included.

Reviewer #1 Comments

1. The manuscript entitled ‘Filling the GAP: Integrating the Gambling Addiction Program into a shelter setting for people experiencing poverty and homelessness’ is a clearly written appraisal of a programme intervention. The rationale for the need to create this intervention programme and details of the programme are explicit; the design and method are clear and sample recruitment is also clearly stated. The programme intervention is a complex one due to the nature of the presenting clientele. The authors acknowledge the intricate systemic web of homelessness, disability, unemployment, and problem gambling and do so with due regard for the dignity of the sample. It is interesting to note that close to two-thirds of the interviewees had post-secondary qualifications. Given the small sample, not much else can be said but it is an interesting bit of information and gambling addiction case workers can tailor their interventions accordingly. Verbatim extracts showcase the verbal ability of some of the clients and given the general level of education among the sample, perhaps the programme can be tailored even more. The iterative and adaptive nature of the programme is to be commended and although it is acknowledged as a potential limitation, it can equally be construed as a necessary adaptation of the programme to ever-changing needs of the clientele. It seems, from the verbatim text, that some clientele found this to be a positive aspect of the programme and further underscored the need to take on board what the clients are saying. The incentive structure is noteworthy where participants received a $10 honorarium for study enrolment, $5 for the call to schedule an interview, and $30 for the interview. The assumption is that participants who fell out of the study part of the way were nonetheless compensated for their time. As soon as housing is secured, attrition rates increase. The latter issue in tandem with the incentive structure could perhaps be looked at a little more skeptically. The range of time spent in the programme is quite large with some participants clearly not involved for any meaningful length of time and others likely benefitting handsomely. The authors do acknowledge that the programme purposely tried to avoid scheduling conflicts (clients receiving benefit payments on days which would have precluded their involvement in the programme). Clearly, the clientele is diverse and given the varied needs, the programme is likely to be more useful to some than others. An interesting finding is the lack of engagement with the financial trustee: “However, low client interest and a long trustee waitlist limited the potential impact of the volunteer trusteeship associated with the GAP”. This is certainly an avenue worth pursuing in terms of increasing engagement with trustees. It is possible that the recent global financial crisis and the reputation of banks has played a role. The fact that participants disengaged from the programme after housing was secured is of concern. Was it felt that programme engagement was perhaps mandatory to securing housing? It may be the case that participants may have felt that engaging with the programme would have increased their chances of securing housing. The authors acknowledge that “addiction recovery is not linear, but rather involves cycles of harm reduction, abstinence, and relapse” and this clearly makes programmes of this nature very complex. Another noteworthy conclusion of the study was the recommendation to improve inter-agency communication and leadership and very importantly to make available shared data – the last aspect is very difficult to address. The complex nature of the issue (homelessness, problem gambling, comorbidity etc) requires a multipronged approach. Sadly, basic things like housing requires political will. Many of the prevalent issues mentioned are the effects of multiple causes from the top down. Nonetheless, programmes of this nature do at least try to grapple with the effects. The research team are to be applauded for their efforts. The assumption with this manuscript is that this is a trial-run of sorts. More data is needed to truly evidence sustained impact.

Response: Thank you for your thoughtful comments on the paper, the program, and the challenges inherent in the lives of the participants. To address your question regarding whether clients felt that program engagement was mandatory to secure housing this is most likely not the case. GSM has been situated in inner city Toronto since 1963. It is widely known as a shelter that offers a wide range of services for people who are experiencing homelessness and it operates from a client-centred approach so that the client decides which challenge(s) are the primary focus of their care. Housing it an inherent aspect of the Good Shepherd service for all clients.

Regarding the level of postsecondary education in the sample. Yes, “Most participants (n = 10) had completed some form of post-secondary education” (p. 11). However, when looking at the larger sample (n = 35) who participated in the study and program, which includes those who did and did not complete an interview, less than half (n = 15 [43%]) had completed some form of post-secondary education—and that is comparable to the level of education data we were able to find (see below). Consistent with research findings that show that people with higher levels of education tend to participate in research interviews, a higher relative proportion of those with postsecondary education completed interviews in our study compared with those who did not complete an interview. We added a footnote to provide greater context and clarity for the reader. (please see more on this in the response #4 for reviewer #3).

Reviewer #2 Comments

1. This paper provides further insight into a meaningful and oft-overlooked area of gambling—problem gambling among those who are homeless or at-risk of becoming homeless. In addition, the paper provides a rich description of the incorporation of a gambling recovery program into a robust multi-service agency. The authors demonstrate that it is possible to successfully incorporate a gambling treatment program into such an environment and that it addresses a significant recovery component for many homeless people. They also provide nuance in describing the challenges of implementing such a program in a real-world clinical setting. The authors do well in describing their process and outcomes and the qualitative approach provides further dimension into the experiences of those struggling with homelessness, PG, and other comorbidities. 

Response: Thank you for your thoughtful comments on the paper and on the nuances of implementing this program in real time and in challenging circumstances.

2. First, please provide information about how many possible participants declined to answer the NODS CLiP. Likewise, the authors state that 33 participants consented to the interview. How many declined?

Response: Each evening Good Shepherd goes through an intake process with men who are seeking overnight shelter. Intake starts at 5:30 in the evening and within 15 minutes the entire process is complete. People are assigned to one of the 95 beds of which 25 are reserved for the Drug and Alcohol Recovery Enrichment Program. This is a fast paced environment where the emphasis is on providing overnight shelter. Men are on the streets all day so they want a bed and shower. It is within this environment that people were asked if they would answer some questions on gambling. Intake staff asked as many of the men as possible if they would like to answer some questions on gambling. If they agreed, these men were taken offside to answer the questions. Refusals could not be captured in this fast-paced environment.

33 people agreed to be contacted for a follow up interview and 28 consented, none declined (they just couldn’t be reached/scheduled).

3. Second, please provide the actual number of men participating in the study. The authors only state that the majority were male and that overall 90% of the GSM are male.

Response: To preserve confidentiality of the few women who participated, we cannot provide the actual percentage of men in the study. We added this information to the manuscript on line 213: (< 6 women).

4. Third, please provide an definition for at-risk of homelessness.

Response: The definition of “At Risk of Houselessness” that is used by Good Shepherd Ministries for client assessment is as follows and included on lines 213-219 of the manuscript: 

The Canadian Definition of Homelessness includes two types of circumstances in the category of those who are “at risk of homelessness:” 

• Precariously Housed: facing serious housing problems, including unaffordable housing, bad housing conditions, overcrowding, or unsafe housing; and 

• At Imminent Risk of Homelessness: facing immediate potential loss of housing due to eviction, inability to pay rent, or violence in the home.

Source: Canadian Homelessness ResearchNetworks, 2012, http://www.homelesshub.ca/ResourceFiles/CHRNhomelessdefinition-¬‐1pager.pdf)

5. Fourth, the authors state that various descriptive statistics were used, but few were reported. It would be informative to have an idea of PG severity among this sample. A count of the number of participants scoring 1, 2, or 3 on the NODS-CLiP and the mean and SD of the PGSI would provide further clinical information of the sample. Please provide this information in the "Description of Study Participants" section.

Response: The mean and standard deviation has been added to the “Description of Study Participants on line 238.” 

As per the response to #2 above we do not have NODS CLiP data

Reviewer #3 Comments

1. Thank you for the opportunity to review this study. This is a very important and understudied topic and I appreciate the care the research team takes in identifying the issues faced by those facing housing instability and poverty. The paper is carefully and clearly written. I have only minor suggestion that may improve the paper.

Response: Thank you for these thoughtful comments regarding the topic and how the issues were addressed.

2. First, I think it would be a good idea to define exactly what you mean by “problem gambling”. Making sure that a clear definition is given so that it does not get conflated with Gambling Disorder or pathological gambling would be a helpful primer for the reader.

Response: The enrolment of clients into the program (and subsequently the study) was based on scoring 3 or more on the Problem Gambling Severity Index. A score of 3 to 7 indicates moderate level of problems leading to some negative consequences and a score of 8 or more indicates problem gambling with negative consequences and a possible loss of control. This information was added to the last paragraph of the section “Gambling Addiction Program (GAP)” in the Methods (please see lines 131-134).

3. Second, I wondered if any consideration was given to the data collection period. I know that challenges for those facing instability in their housing options vary from season to season in a place like Toronto.

Response: We didn’t specifically consider this but the interviews cover a time span of November 2018 to April 2019. The data collection window was selected based on the end of the pilot of the implementation phase of the intervention and with the project end date in mind (March 2019) to ensure time for data analysis and deadline for submission of a report to the funding body. November to April represent our colder months, and it’s possible that shelters in the city operate at higher capacity thus creating greater opportunity for people to hear about the program. 

4. The level of some post-secondary education seemed quite high in the client sample. I would be interested to see how this relates to existing literature/estimates for homeless populations and whether it might impact study/program participation.

Response: Good point. Yes, “Most participants (n = 10) had completed some form of post-secondary education” (p. 11). However, when looking at the larger sample (n = 35) who participated in the study and program, which includes those who did and did not complete an interview, less than half (n = 15 [43%]) had completed some form of post-secondary education—and that is comparable to the level of education data we were able to find (see below). Consistent with research findings that show that people with higher levels of education tend to participate in research interviews, a higher relative proportion of those with postsecondary education completed interviews in our study compared with those who did not complete an interview. We added a footnote to provide greater context and clarity for the reader (please see page 11).

There is one recent paper by Claveau (2020) that reports on housing need and education using The Canadian Housing Survey 2018 (https://www150.statcan.gc.ca/n1/pub/75f0002m/75f0002m2020003-eng.htm). She does not cross education with homelessness, however. The percentage distribution of persons in households by highest level of education is instructive, for the group experiencing the greatest housing precarity (i.e., those closest to being at risk, if not currently at risk, of homelessness—renters of social and affordable housing) had comparable levels of education:

Percentage distribution of persons in households by highest level of education: 39% of renter households in social and affordable housing have post-secondary education (post-secondary diploma or certificate 23% + university degree 16%). 

Similarly, a 2016 Statistics Canada report on hidden homelessness in Canada—where “Hidden homelessness is defined as those who ever had to temporarily live with family, friends or in their car because they had nowhere else to live” (Rodrigue 2016: 9)—also reports comparable levels of education. The proportion of Canadians aged 15 and over who experienced hidden homelessness and who completed some post-secondary education was 40.2% (Trade certificate or diploma 10.1%, College CEGEP or other certificate or diploma 9.4%, university degree below bachelor’s 7.9%, Bachelor’s degree 7%, University degree above bachelor’s 5.8%). (https://www150.statcan.gc.ca/n1/pub/75-006-x/2016001/article/14678-eng.htm) 

On another note, I am not sure how much national data will help in explaining the higher rates of education among our sample. The population we are looking at is not representative of homeless people in Canada given that it is a small convenience sample located in the City of Toronto. Other data and literature is needed to shed some light what might be going on in Toronto, the site of our study. The 2016 Census data indicates that Toronto has a higher percentage of those with a bachelor degree or higher compared to Canada as a whole (40.9% vs 28.5%) (https://www12.statcan.gc.ca/census-recensement/2016/as-sa/fogs-spg/Facts-cma-eng.cfm?LANG=Eng&GK=CMA&GC=535&TOPIC=10). 

At the provincial level, there is an ongoing housing crisis in Ontario and rising housing costs as well as other larger structural issues that have increased homelessness, including the global recession and neo-liberalism in Canada. We acknowledge the literature still tends to find that lower education is associated with homelessness, but with a changing job market, rising precarity in jobs, and few basic economic supports, alongside rising rents and cost of living, perhaps education can only do so much to safeguard against homelessness. Finally, there is also the issue of ambling and it not necessarily being an activity engaged in by people of lower SES, but still having the potential to create great hardship in the lives of those who gamble at the problematic level. 

https://digitalscholarship.tsu.edu/jpmsp/vol24/iss1/6/

https://www.homelesshub.ca/resource/where-will-we-live-ontarios-affordable-housing-crisis

http://universitypublications.net/ijas/0705/pdf/P4RS226.pdf

5. Finally, in several cases the relevance of the excerpts to the themes being discussed were presented as self-evident. It would be helpful to the reader for the writing team to make a clearer connection between the theme and the evidence of that theme. One example is the excerpt at the top of page 15 (case 304). An explanation of the variability of content and its importance to keeping this client engaged would make the connection of evidence to theme more explicit and do more to demonstrate the value of a tailored approach.

Response: We appreciate the careful reading and consideration. In part, this is a question of style in the reporting and presentation of qualitative research, but still the reviewer has the reader in mind, and makes an excellent suggestion to be more explicit in connecting analytical themes and illustrations to enhance the authors’ case and readers’ comprehension.

In addition to addressing the specific example of case 304, we’ve reviewed and revised throughout the results accordingly, attending to the relationship and fit between analytical themes, illustrations, and commentary.

6. Repetition of integration in abstract “The integration of gambling treatment into this multiservice delivery model addressed the complex needs of the service users through integrated and person-centered approaches”

Response: Thank you the sentence has been changed to: “The introduction of gambling treatment into this multi-service delivery model addressed the complex needs of the service users through integrated and person-centered approaches to care that responded to client needs, fostered therapeutic relationships, reduced experiences of discrimination and stigma, and enhanced recovery.”

7. Progrom: change to program.

Response: Thank you. This has been corrected.

8. Pg 3: well positioned= well-positioned

Response: Thank you. This has been corrected.

9. The acronym GSM is used before it is defined

Response: The first occurrence of GSM is on page 3 and introduced with the full name of the agency. We did notice that the acronym GAP was used in the abstract without being defined and have made this change.

Page 5: “GACWs performed outreach at other community agencies serving the target population, which provided the basis for external referrals to the program.” Change “,which” to “that”. The use of which instead of that happens a lot throughout the manuscript

Response: Thank you for catching that one--revised. We further found and reviewed 16 additional examples of ostensibly non-restrictive clauses set off by “, which…” throughout the paper and revised for clarity, grammar, and style.

---

## [Decision Letter · Decision Letter 1]

21 Feb 2022

Filling the GAP: Integrating the Gambling Addiction Program into a shelter setting for people experiencing poverty and homelessness

PONE-D-21-07198R1

Dear Dr. Matheson,

We’re pleased to inform you that your manuscript has been judged scientifically suitable for publication and will be formally accepted for publication once it meets all outstanding technical requirements.

Kind regards,

Marc Potenza

Academic Editor

PLOS ONE

Additional Editor Comments (optional):

Reviewers' comments:

Reviewer's Responses to Questions

**Comments to the Author**

1. If the authors have adequately addressed your comments raised in a previous round of review and you feel that this manuscript is now acceptable for publication, you may indicate that here to bypass the “Comments to the Author” section, enter your conflict of interest statement in the “Confidential to Editor” section, and submit your "Accept" recommendation.

Reviewer #1: All comments have been addressed

Reviewer #3: All comments have been addressed

2. Is the manuscript technically sound, and do the data support the conclusions?

Reviewer #1: Yes

Reviewer #3: Yes

3. Has the statistical analysis been performed appropriately and rigorously? 

Reviewer #1: N/A

Reviewer #3: N/A

4. Have the authors made all data underlying the findings in their manuscript fully available?

Reviewer #1: No

Reviewer #3: Yes

5. Is the manuscript presented in an intelligible fashion and written in standard English?

Reviewer #1: Yes

Reviewer #3: Yes

6. Review Comments to the Author

Reviewer #1: The authors have attended to the queries raised and have provided very detailed contextual information on the current housing crisis in Canada. In fact, some of the data cited is very interesting. Until quite recently, homelessness was considered as affecting only less educated people. However, this is changing. In Ireland, the homelessness situation is also starting to reflect a more diverse grouping of people. Several cities across the developed world are showing the same thing. Anyway, I think the paper is very good and should be published. Best, Raegan

Reviewer #3: Thank you for incorporating my suggestions and doing this important work!---------------------------

7. PLOS authors have the option to publish the peer review history of their article (what does this mean?). If published, this will include your full peer review and any attached files.

Reviewer #1: **Yes: **Raegan Murphy

Reviewer #3: **Yes: **Mark van der Maas

---

## [Editor Report · Acceptance letter]

1 Mar 2022

PONE-D-21-07198R1 

Filling the GAP: Integrating a gambling addiction program into a shelter setting for people experiencing poverty and homelessness 

Dear Dr. Matheson:

I'm pleased to inform you that your manuscript has been deemed suitable for publication in PLOS ONE. Congratulations! Your manuscript is now with our production department. 

Kind regards, 

on behalf of

Dr. Marc Potenza 

Academic Editor

PLOS ONE